# The matrix vesicle cargo miR-125b accumulates in the bone matrix, inhibiting bone resorption in mice

Tomoko Minamizaki[1], Yuko Nakao[1,2], Yasumasa Irie[1,3], Faisal Ahmed[1], Shota Itoh[1,2], Nushrat Sarmin[1], Hirotaka Yoshioka [1,8], Asako Nobukiyo[4], Chise Fujimoto[1], Shumpei Niida[5], Yusuke Sotomaru[4], Kotaro Tanimoto[2], Katsuyuki Kozai[3], Toshie Sugiyama[6], Edith Bonnelye[7], Yuichiro Takei [1,9] & Yuji Yoshiko[1]*

Communication between osteoblasts and osteoclasts plays a key role in bone metabolism. We describe here an unexpected role for matrix vesicles (MVs), which bud from bone-forming osteoblasts and have a well-established role in initiation of bone mineralization, in osteoclastogenesis. We show that the MV cargo miR-125b accumulates in the bone matrix, with increased accumulation in transgenic (Tg) mice overexpressing miR-125b in osteoblasts. Bone formation and osteoblasts in Tg mice are normal, but the number of bone-resorbing osteoclasts is reduced, leading to higher trabecular bone mass. miR-125b in the bone matrix targets and degrades *Prdm1*, a transcriptional repressor of anti-osteoclastogenic factors, in osteoclast precursors. Overexpressing miR-125b in osteoblasts abrogates bone loss in different mouse models. Our results show that the MV cargo miR-125b is a regulatory element of osteoblast-osteoclast communication, and that bone matrix provides extracellular storage of miR-125b that is functionally active in bone resorption.

[1] Department of Calcified Tissue Biology, Graduate School of Biomedical and Health Sciences, Hiroshima University, 1-2-3 Kasumi, Minami-ku, Hiroshima 734-8553, Japan. [2] Department of Orthodontics and Craniofacial Developmental Biology, Graduate School of Biomedical and Health Sciences, Hiroshima University, 1-2-3 Kasumi, Minami-ku, Hiroshima 734-8553, Japan. [3] Department of Pediatric Dentistry, Graduate School of Biomedical and Health Sciences, Hiroshima University, 1-2-3 Kasumi, Minami-ku, Hiroshima 734-8553, Japan. [4] Natural Science Center of Basic Research and Development, Hiroshima University, 1-2-3 Kasumi, Minami-ku, Hiroshima 734-8551, Japan. [5] Medical Genome Center, National Center for Geriatrics and Gerontology, 7-430 Morioka-cho, Obu 474-8551, Japan. [6] Department of Animal Science, Graduate School of Science and Technology, Niigata University, 2-8050 Ikarashi, Nishi-ku, Niigata 950-2181, Japan. [7] CNRS ERL 6001/ INSERM U1232, Institut de Cancérologie de l'Quest, 44805 Saint-Herblain, France. [8] Present address: Department of Anatomy, School of Medicine, International University of Health and Welfare, 4-3 Kozunomori, Narita 286-8686, Japan. [9] Present address: Department of Nutrition, Faculty of Nutrition, University of Kochi, 2751-1 Ike, Kochi 781-8515, Japan. *email: yyuji@hiroshima-u.ac.jp

Cell-to-cell communication is pivotal for all multicellular organisms. Most cell types exchange information through soluble factors or direct interaction, but they also release extracellular vesicles (EVs) that can have an impact on neighboring or distant cells. The biological significance of EVs has been affirmed in numerous studies on intercellular communication that document the transfer of proteins, lipids, and nucleic acids; of these, microRNAs (miRNAs) have drawn much attention as EV cargos[1,2]. miRNAs, ~22 nucleotides in length, constitute a class of non-coding RNAs and induce gene silencing by binding to target sites within the 3′UTR of targeted mRNAs, resulting in suppressing protein synthesis and/or mRNA degradation[3]. miRNAs have been shown to be involved in a wide range of biological processes such as cell cycle control[4], developmental processes[5], glucose homeostasis[6], immune responses[7], bone metabolism[8], and so on.

In general, EVs are released into the extracellular fluids; however, matrix vesicles (MVs) budding from polarized mature osteoblasts accumulate in the unmineralized bone matrix (osteoid). MVs in the osteoid initiate calcium and phosphate crystal formation inside vesicles by utilizing the calcium channels, phosphate transporters, and enzyme activities required in calcium and phosphate transport and initial mineral formation. Calcium phosphate crystals thereafter grow, penetrate the lipid bilayer of MVs, and expand to mineralize bone matrix[9]. Advances in global analysis indicate that MVs share certain features in common with EVs[10,11], including the presence of miRNAs[12] and of numerous proteins (see http://bioinf.xmu.edu.cn/MVsCarta/), some of which are known regulators of osteoclast and osteoblast development (e.g., TNFRSF11B[13], WNT5A[14], SEMA4D[15]), but whether they are functionally active when carried as cargo is not yet known. We therefore tested the hypothesis that MVs play an important, previously undefined role in osteoclasts and osteoblasts beyond their known involvement in bone matrix mineralization. We report here that miR-125b is a MV cargo that accumulates in bone matrix where it inhibits osteoclastogenesis and bone resorption via downregulation of the transcriptional repressor PRDM1.

## Results

**MVs inhibit osteoclast formation in vitro.** To address possible novel bone biological functions of MVs, we isolated MVs from the extracellular matrix (ECM) deposited by mouse osteoblastic MC3T3-E1 cells. The isolated vesicles fulfilled the expected morphological and biochemical criteria of MVs[9] (Supplementary Fig. 1a–d). Treatment of mouse bone marrow macrophage (BMM) cultures with isolated MVs in the presence of CSF1 (M-CSF) and TNFSF11 (RANKL)[16–18] decreased formation of tartrate-resistant acid phosphatase (TRAP)-positive multinucleated cells (MNCs, i.e., osteoclasts) (Fig. 1a); no osteoclasts formed with or without MVs in the absence of RANKL (Fig. 1a). Consistent with the histochemical results, treatment with MVs also decreased the expression of osteoclast marker genes *Nfatc1*, a transcription factor necessary for osteoclastogenesis[18], *Dcstamp*[19] and *Ctsk*[20] (Fig. 1b) as well as resorption pit formation[21] (Fig. 1c). Similar effects were observed in the mouse macrophage cell line, RAW-D (Supplementary Fig. 2a, b) with RANKL, in which an accompanying uptake of dye from labeled MVs to cells was also evident (Supplementary Fig. 2c). Treatment with MVs from rat and human primary osteoblast cultures (Supplementary Fig. 2d, e) confirmed the presence of anti-osteoclastogenic factor(s) in MVs across species. The RANKL-decoy receptor TNFRSF11B (osteoprotegerin (OPG)), which inhibits osteoclastogenesis[13,22], has been reported in MVs in the conditioned medium of vascular smooth muscle cells (http://bioinf.xmu.edu.cn/MVsCarta/).

Contrary to smooth muscle cells, we did not detect OPG in our osteoblast-derived MVs (Supplementary Fig. 1c), suggesting that MVs inhibit osteoclast formation independently of the RANKL-OPG pathway. Although dye transfer occurred from labeled MVs to MC3T3-E1 cells (Supplementary Fig. 3a), MVs did not elicit significant effects on osteogenic differentiation and bone-like nodule formation in these (Supplementary Fig. 3b) or rat primary osteoblast (Supplementary Fig. 3c) cultures. The data suggest that MVs modulate osteoclastogenesis but not osteoblastogenesis in vitro.

**miR-125b mimics the MV effects in vitro.** Recently PCR array profiling has identified miRNAs in MVs isolated from rat chondrocyte cultures[12]. We therefore used miRNA microarray analysis of osteoblastic (MC3T3-E1) MVs and identified 176 miRNAs (GSE140242 at NCBI GEO), 76 of which were common between mouse and human (Supplementary Data 2), with several good candidates for regulators of genes involved in osteoclast formation[23–27] (Supplementary Table 1). Of these, miR-125b was more abundant in MVs than in the parent MC3T3-E1 cells from which they derived (Fig. 1d), primary human and rat osteoblasts (Fig. 1e) and RAW-D cells cultured with or without RANKL (Fig. 1f). Notably, however, treatment with MVs increased miR-125b levels in RAW-D cells (Fig. 1g). We then established a subclone of MC3T3-E1 cells with lower levels of miR-125b (Supplementary Fig. 4a) but normal osteogenic capacity (E1[low]) (Supplementary Fig. 4b) to do a loss-of-function study. MVs from E1[low] cells had no significant effect on TRAP+ MNC formation in RAW-D cells (Supplementary Fig. 4c). Meanwhile, transfection of miR-125b mimic (Fig. 1g) had higher efficacy than those with control miRNA against TRAP+ MNC formation (Fig. 1h) and associated gene expression (*Nfatc1*, *Dcstamp*, and *Catsk*) (Fig. 1i). miR-125b mimic (Supplementary Fig. 4d) did not affect osteogenesis in MC3T3-E1 or the mouse mesenchymal stem cell line ST2 models (Supplementary Fig. 4e, f). These data indicate that miR-125b participates in the inhibitory effect of MVs on osteoclastogenesis in vitro.

**Transgenic mice have high bone mass and fewer osteoclasts.** To determine whether miR-125b as MV cargo plays a functional role in bone in vivo, we generated transgenic (Tg) mouse lines (#5, #7, and #16; Supplementary Fig. 5a) expressing mmu-miR-125b-5p in osteoblasts by using human osteocalcin promoter[27] (Fig. 2a, b). Mice (males) of the three lines at 9- and 12-weeks of age exhibited identical phenotypes including high bone mass (Supplementary Fig. 5b); line #5 was used for further analyses. Figure 2b shows cell- and tissue-specific expression of miR-125b in Tg mice, which were born at normal Mendelian ratios and grew normally (Supplementary Fig. 5c). Tg femurs (12-week-old males) and vertebrae (L3 of 10-week-old females) exhibited higher bone mass than their corresponding wild-type (WT) mice (Fig. 2c), due to increased trabecular bone (12-week-old females, Fig. 2d); histological assessment of Tg tibiae (9- and 10-week-old female mice) also confirmed this event (Supplementary Fig. 6a–c). WT and Tg mice displayed equal numbers of alkaline phosphatase (ALP)-positive osteoblasts (Fig. 2e) and equal mineral apposition rates (MARs; calcein double-labeling) (Fig. 2f), indicating that bone formation was not affected by miR-125b overexpression in osteoblasts. However, the percentage of TRAP+ cell surface per trabecular bone surface was lower in Tg compared to WT mice (Fig. 2g). No difference in TRAP staining was seen in the zone of calcified cartilage and subchondral bone (Fig. 2h), which may reflect lower accumulation of miR-125b in bone at the chondro-osseous junction in Tg mice.

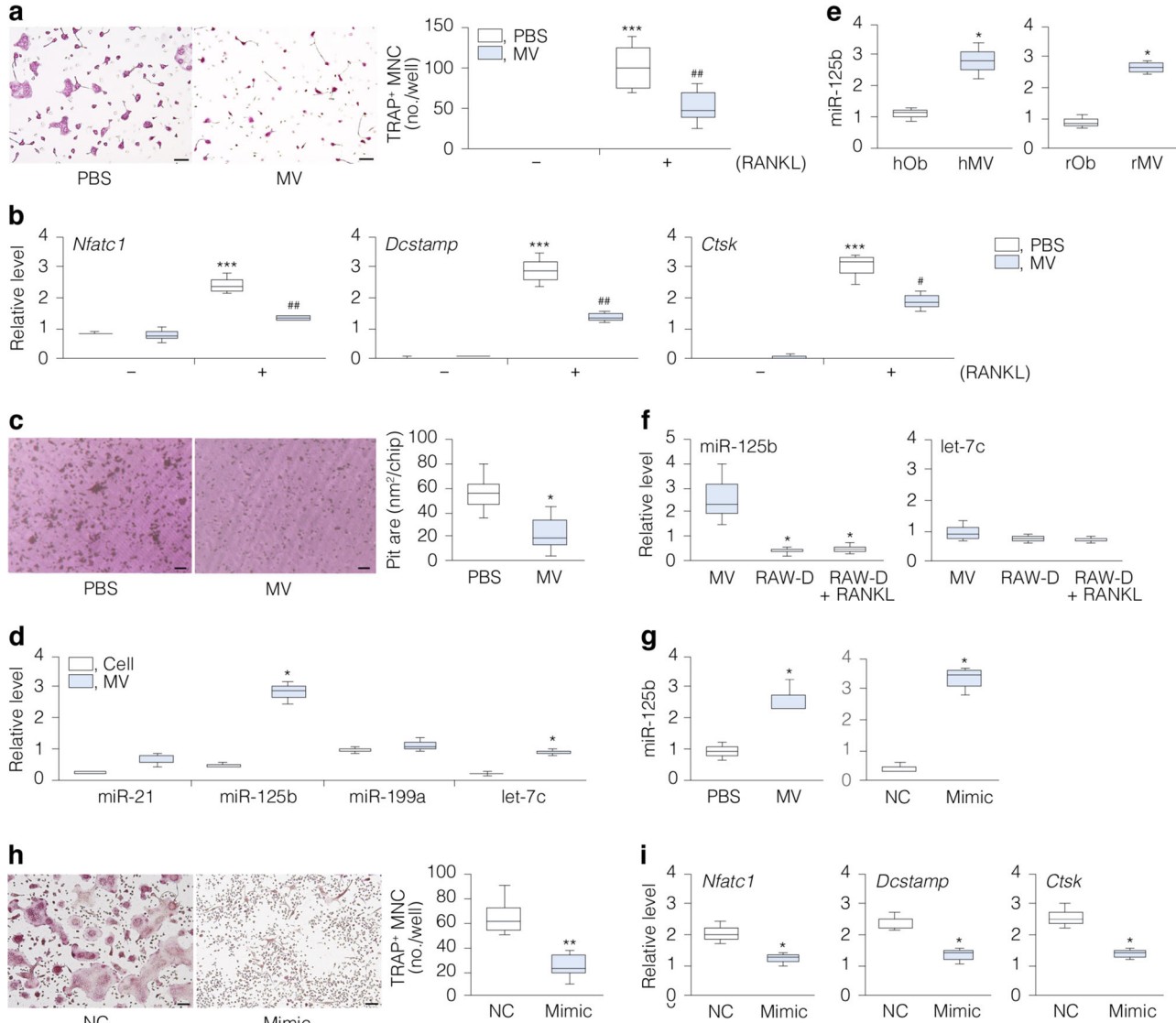

**Fig. 1 MVs and miR-125b impair TRAP⁺ multinucleated cell (MNC) formation.** To induce TRAP⁺ MNC formation, BMMs were cultured in the presence of M-CSF alone or M-CSF plus RANKL (**a**–**c**). Cells were also treated with MVs (1 μg protein/mL) or PBS. **a** Representative images of TRAP⁺ MNCs (with M-CSF and RANKL; scale bars, 100 μm) and the number of TRAP⁺ MNCs/well in the presence of MVs or PBS ($n = 6$). **b** Expression of osteoclast marker genes (*Nfatc1*, *Dcstamp*, and *Ctsk*); *Rpl32* was used as internal control ($n = 3$). **c** Representative images of resorption pits (scale bars, 50 μm) and pit areas on dentin slices stained with hematoxylin ($n = 6$). **d** miRNA levels in MVs vs. their source MC3T3-E1 cells ($n = 3$). **e** miR-125b levels in human and rat osteoblasts (hOb and rOb) and their MVs (hMV and rMV) ($n = 3$). Cells were cultured until osteoid-like nodules formed (**d**, **e**). To induce TRAP⁺ MNC formation, RAW-D cells were treated with or without RANKL (**f**–**i**). **f** miR-125b and let-7c levels in MVs vs. RAW-D cells ($n = 3$). **g** miR-125b levels in RAW-D cells treated with MVs or PBS, or transfected with miR-125b mimic (Mimic) or negative control miRNA (NC) ($n = 3$). U6 was used as internal control (**d**–**g**). **h** Representative images of TRAP⁺ MNCs (scale bars, 100 μm) and the number of TRAP⁺ MNCs/well in the presence of Mimic or NC ($n = 6$). **i** Expression of osteoclast marker genes (*Nfatc1*, *Dcstamp*, and *Ctsk*) in RAW-D cells with Mimic or NC ($n = 3$); *Rpl32* was used as internal control. *, #$P < 0.05$, **, ##$P < 0.01$, and ***$P < 0.001$ vs. matched control by Tukey's multiple comparison (**a**, **b**, **f**) and by Student's *t*-test (**c**–**e**, **g**–**i**).

**miR-125b targets *Prdm1* in osteoclast precursors**. To identify the mechanisms underlying MV-mediated impaired osteoclastogenesis, we analyzed Tg and WT bone and bone marrow populations ex vivo. No difference was observed in bone formation between Tg and WT bone marrow stromal cell cultures (Fig. 3a) or between Tg and WT calvaria cell cultures (Supplementary Fig. 7a). Similarly, no difference was seen between Tg and WT cellular and MV ALP activity (Supplementary Fig. 7b) or their ALPL and ANXA5 levels (Supplementary Fig. 7c). However, higher levels of miR-125b were present in Tg vs. WT MVs (Supplementary Fig. 7d) and in Tg vs. WT bone matrix (Fig. 3b). Given that miR-125b was not detected in conditioned media from either Tg or WT calvaria cell cultures (Supplementary Fig. 7d),

these results suggest that miR-125b secreted from osteoblasts is mostly or all sequestered in the bone matrix via MVs.

Neither TRAP⁺ MNC formation (Fig. 3c) nor ability to resorb bone in vitro (Supplementary Fig. 7e) was different between Tg and WT BMM cell cultures. Similarly, and consistent with the view that osteoblast-secreted miR-125b is undetectable in conditioned medium but is sequestered in the bone matrix via MVs, osteoclast formation was the same in Tg and WT BMMs cocultured with either Tg or WT osteoblasts (calvaria cells forming osteoid-like nodules) (Supplementary Fig. 7f). FACS analysis revealed that CD117⁺/CD115⁺/CD11b^low osteoclast progenitors[28] (Fig. 3d), macrophages, hematopoietic stem cells, and B cells were comparable between the two strains, with a small but substantial reduction

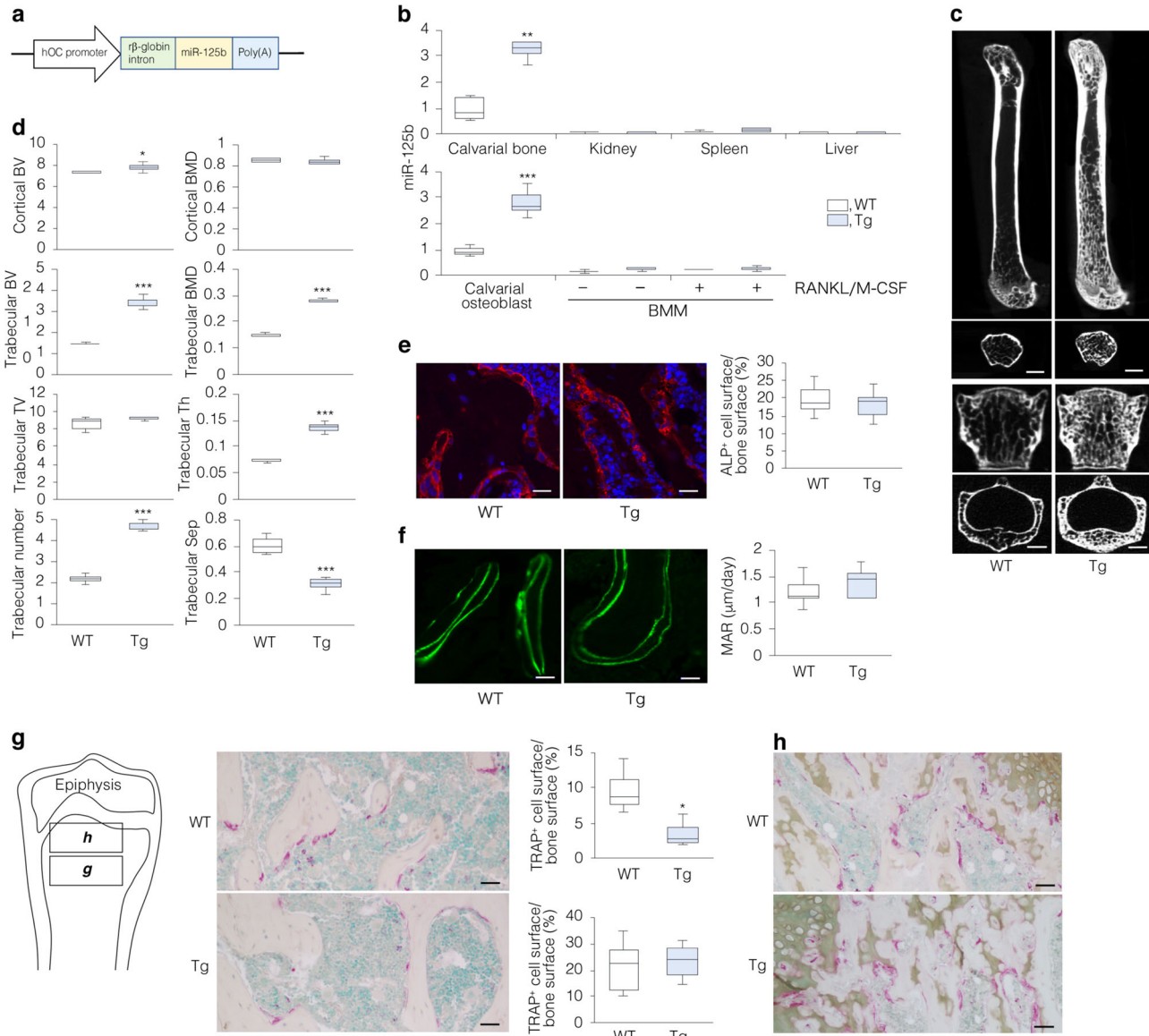

**Fig. 2 Transgenic (Tg) mice overexpressing miR-125b in osteoblasts exhibit a high bone mass phenotype. a** Transgene construct. hOC, human osteocalcin promoter; rβ-globin intron, rabbit β-globin intron. **b** miR-125b levels in various tissues and cells of 12-week-old male transgenic (Tg) and wild-type (WT) mice; U6 was used as internal control (*n* = 3). **c** Microcomputed tomography (μCT) of 12-week-old male femurs (upper panels) and 10-week-old female lumber vertebrae (L3, lower panels) (representative images of each phonotype; upper and lower panels indicate longitudinal and transverse views, respectively). Scale bars, 1 mm (femurs) and 0.5 mm (L3). **d** Bone morphometric parameters of 12-week-old Tg and WT female distal femurs (WT, *n* = 5; Tg, *n* = 7). Analysis was performed on the metaphyseal region with 7 μm width, based on the position 1 μm away from the distal femoral growth plate. BV, bone volume (mm$^3$); BMD, bone mineral density (mg/mm$^3$); TV, tissue volume (mm$^3$); Th, thickness (mm); Trabecular number (no./mm); Sep, separation (mm). **e**–**h** Histology of 10-week-old Tg and WT female distal femurs (representative images in each genotype, *n* = 5). **e** ALPL-immunoreactive cells delineate osteoblasts and the percentage of ALP$^+$ cell surface per bone surface. Nuclear counterstaining with DAPI (scale bars, 25 μm). **f** Calcein double-labeling (scale bars, 20 μm) and the corresponding mineral apposition rate (MAR) (*n* = 5). **g**, **h** TRAP$^+$ multinuclear cells (MNCs) delineate osteoclasts (scale bars, 50 μm) and the percentage of TRAP$^+$ cell surface per bone surface. Left panel in **g** shows schematic image of the distal femur at low magnification; cells within the square frames **g** and **h** were counted and plotted in **g** and **h**, respectively. Nuclear counterstaining with methyl green. *$P < 0.05$, **$P < 0.01$, and ***$P < 0.001$ vs. WT by Student's *t*-test.

in T cells in Tg mice in comparison with those in WT mice (Supplementary Fig. 8). Taken together, these data suggest that miR-125b trapped in bone matrix inhibits osteoclastogenesis. Indeed, treatment of RWA-D cells with Tg MVs resulted in greater inhibition of osteoclast formation than that with WT MVs (Fig. 3e). When either Tg or WT BMMs were cultured on Tg vs. WT bones in the presence of RANKL and M-CSF, TRAP$^+$ spots were less on Tg than WT bones (Fig. 3f). These results suggest that bone matrix miR-125b decreases osteoclastogenesis.

We then searched public databases to identify putative miR-125b target(s) (Supplementary Table 2). Of these, PR domain containing 1, with zinc finger domain (PRDM1), a transcriptional repressor of anti-osteoclastogenic factors[16], was selectively downregulated in RAW-D cells transfected with miR-125b mimic (Supplementary Fig. 9) or treated with MVs (Fig. 3g). Luciferase activity was reduced in RAW-D cells co-transfected with miR-125b mimic and *Prdm1* 3′UTR reporter vector, but not with miR-125b mimic and *Prdm1* mutant 3′

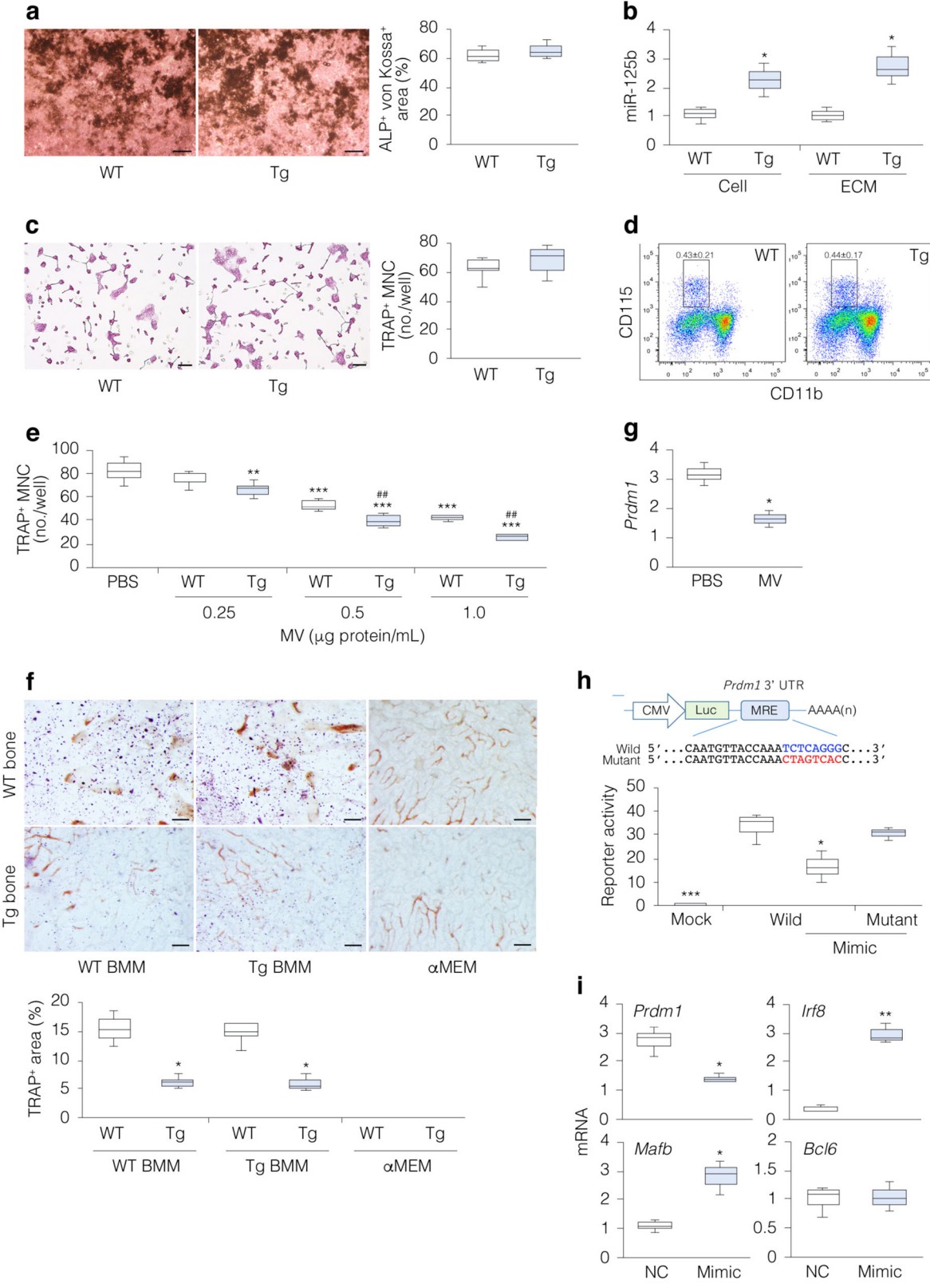

UTR reporter vector[29] (Fig. 3h). Further, the miR-125b mimic decreased levels of *Prdm1* in RAW-D cells and concomitantly increased mRNA (Fig. 3i) and protein (Supplementary Fig. 10) levels of two known downstream targets of PRDM1 that are inhibitors of osteoclastogenesis, interferon regulatory factor 8 (*Irf8*)[30] and v-maf musculoaponeurotic fibrosarcoma oncogene homolog B (*Mafb*)[31], but not of a third potential inhibitor, B-cell lymphoma 6 (*Bcl6*)[16]. The latter may reflect the fact that we assessed at day 4 and may have missed later events in osteoclastogenesis, as it is known that *Bcl6* is downregulated at a later stage of osteoclastogenesis (by 8 days)[16] than either *Irf8*[30] or *Mafb*[31](by 24 h). Nevertheless, our results strongly suggest that miR-125b suppresses osteoclast formation via downregulation of PRDM1.

**Fig. 3 miR-125b in bone matrix inhibits osteoclastogenesis by targeting *Prdm1*. a** Representative images of osteogenic activities in Tg and WT bone marrow cell cultures, assessed by ALP/von Kossa staining (scale bars, 4.5 mm) and the percentage of ALP/von Kossa+ areas (n = 3). **b** miR-125b levels in AGO2 immunoprecipitates from the cell and ECM fractions of WT vs. Tg bones (n = 3). **c** Representative images of TRAP+ multinucleated cells (MNCs) in WT and Tg bone marrow macrophage (BMM) cultures stimulated with RANKL and M-CSF (scale bars, 100 μm) and the number of TRAP+ MNCs/well (n = 6). **d** FACS analysis of Tg and WT osteoclast precursor (CD117+/CD115+/CD11b low) cells; cells were gated on CD117 (n = 8). **e** The number of TRAP+ MNCs/well in RAW-D cell cultures with or without WT MVs vs. Tg MVs (n = 6). Cells were stimulated with RANKL. **f** Representative images of TRAP staining of either Tg or WT BMMs on Tg vs. WT calvarial bone chips (scale bars, 100 μm) and the percentage of TRAP+ areas (n = 4). αMEM shows bone chips without BMMs. **g** *Prdm1* levels in RAW-D cells with MVs (1 μg protein/mL) or PBS; *Rpl32* was used as internal control (n = 3). **h** The reporter constructs contained the miRNA response element (MRE) (Wild) of *Prdm1* or its mutant, and reporter activities in RAW-D cells cotransfected with reporter plasmids and miR-125b mimic (Mimic) (n = 4). **i** mRNA levels of *Prdm1* and associated downstream target genes in RAW-D cells transfected with Mimic and negative control miRNA (NC); *Rpl32* was used as internal control (n = 3). BMMs and RAW-D cells were stimulated by RANKL plus M-CSF (**c**, **f**) and RANKL (**e**, **g**–**i**), respectively. *P < 0.05, **, ##P < 0.01, and ***P < 0.001 vs. matched control by Student's *t*-test (**b**, **f**, **g**, **i**) and by Tukey's multiple comparison (**e**, **h**).

**miR-125b protects bone loss in mouse disease models**. To determine whether MV-based transfer of miR-125b to bone matrix represents a potential therapeutic target, we ovariectomized (OVX) 8-week-old female mice, as a model of postmenopausal osteoporosis[32]. Four weeks post-OVX, body weight was increased (Fig. 4a) and uteri were hypoplastic (Fig. 4b) in both WT and Tg mice, indicating effective estrogen depletion. However, OVX-induced trabecular bone loss was lower in Tg compared to WT femora (Fig. 4c–e). Trabecular bone loss was also markedly blunted in 10-week-old male Tg vs. WT mice subjected to sciatic neurectomy (NX), a model of disuse or immobilization-induced osteoporosis[33] (Supplementary Fig. 11a, b). Thus, overexpression of miR-125b in osteoblasts confers resistance to bone loss in mouse osteoporosis models. Finally, to confirm that miR-125b inhibits osteoclastogenesis in vivo, we used subcutaneous injections of bacterial lipopolysaccharides (LPSs) over the calvariae to increase osteoclastogenesis and induce osteolysis[30]. Whole-mount staining showed that LPS-induced TRAP staining was attenuated, when miR-125b mimic was co-injected subcutaneously (Fig. 4f). These results indicate that miR-125b can inhibit bone resorption in at least some pathologic conditions. Taken together, our data identify miR-125b accumulated in bone matrix as a player in bone metabolism via osteoblast–osteoclast coupling and as a potential therapeutic target to abrogate bone loss.

## Discussion

MVs budding from osteoblasts are known to accumulate in osteoid and play a crucial role in the initial steps of bone mineralization[9]. However, potential but undefined roles of MVs in bone metabolism have been postulated recently, based on comprehensive analyses of proteins in MVs[10,11]. Together with other recent findings of miRNAs as functional EV cargos[1,2], these prompted us to test for functional roles of MVs in bone. We report here that miRNAs in osteoblasts are transported to bone matrix by MVs. Of these, miR-125b is selectively and abundantly incorporated into MVs and stored in bone matrix. During bone resorption, miR-125b is released and targets PRDM1, a transcriptional repressor of anti-osteoclastogenic factors[16,17] in osteoclast precursors. Tg mice overexpressing miR-125b in osteoblasts exhibit increased miR-125b levels in MVs and bone matrix, resulting in decreased osteoclast numbers and increased bone mass. Overexpressing miR-125b in osteoblasts also abrogates the bone loss associated with estrogen deficiency and immobilization in mice, consistent with the suppression of LPS-induced osteoclast formation and osteolysis by injecting mice with miR-125b mimic.

That osteoblasts secrete EVs and/or MVs with active biological molecules as cargo is not in itself surprising. Indeed, a list of numerous cargo proteins is available at http://bioinf.xmu.edu.cn/ MVsCarta/, but little is known about whether MV cargo molecules accumulate in bone matrix, or have a substantial impact on bone metabolism or turnover. Treatment of mouse BMMs with EVs isolated from conditioned media of primary mouse osteoblast cultures has been shown to promote survival of osteoclasts by the action of the cargo RANKL[34]. However, we found that MVs do not mimic this effect, but rather inhibit formation of osteoclasts and resorption pits, suggesting that anti-osteoclastogenic factor(s) accumulate in MVs and are transported to bone matrix via MVs but not EVs. Recently, MVs isolated from rat growth plate chondrocytes were reported to be selectively enriched in certain miRNAs, including miR-122-5p, miR-142-3p, miR-223-3p, and miR-451-5p[12]. We then focused our attention on miRNAs as MV cargos. Interestingly, the enrichment profile we identified in mouse, rat, and human osteoblast MVs is distinctly different from that in rat chondrocytes[12], suggesting that MVs from at least certain types of chondrocytes exhibit a functionally distinct role, from osteoblast MVs.

By in silico analyses, qRT-PCR screens, and selection of candidate miRNAs more abundant in osteoblasts and MVs than osteoclasts or their precursor cells (BMMs and RAW-D cells), we identified miR-125b for further study. We confirmed that the MV cargo miR-125b regulates osteoclastogenesis but not osteoblastogenesis by gain- and loss-of function experiments in vitro. While overexpression of miR-125b in osteoblasts clearly decreases osteoclastogenesis, we found no evidence for miR-125b regulation of osteoblast differentiation or bone formation in vivo. Of previous studies assessing miR-125b in osteoblast differentiation, only two have reported a regulatory role for miR-125b in osteogenesis, one with human bone marrow stromal cells, which are from a multipotent stem cell-like population differentiated under the influence of a medium whose constituents were undisclosed or transplanted on demineralized bone matrix[35], and the other with ST2 stem-like cells in the presence of BMP-4[36]. We have found no detectable effect of miR-125b on the MC3T3-E1 model cultured under osteogenic conditions or ST2 cells undergoing osteogenesis in vitro in the absence of BMP-4. We also found no effect on osteogenesis when we overexpressed miR-125b in post-mitotic mature osteoblasts ex vivo and in vivo. It is not surprising that miR-125b effects are different in different conditions in vitro and in vivo, given the likelihood of extracellular signal- and/or differentiation stage-specific miR-125b target gene expression profiles in these different models. It is also possible that the ability to detect effects of exogenously added miR-125b on osteoblast development in vitro and ex vivo/in vivo are abrogated by the already abundant expression of miR-125b in these cells.

It is known that circulating miRNAs, whether within EVs or not, are mostly associated with RNA-induced silencing complexes

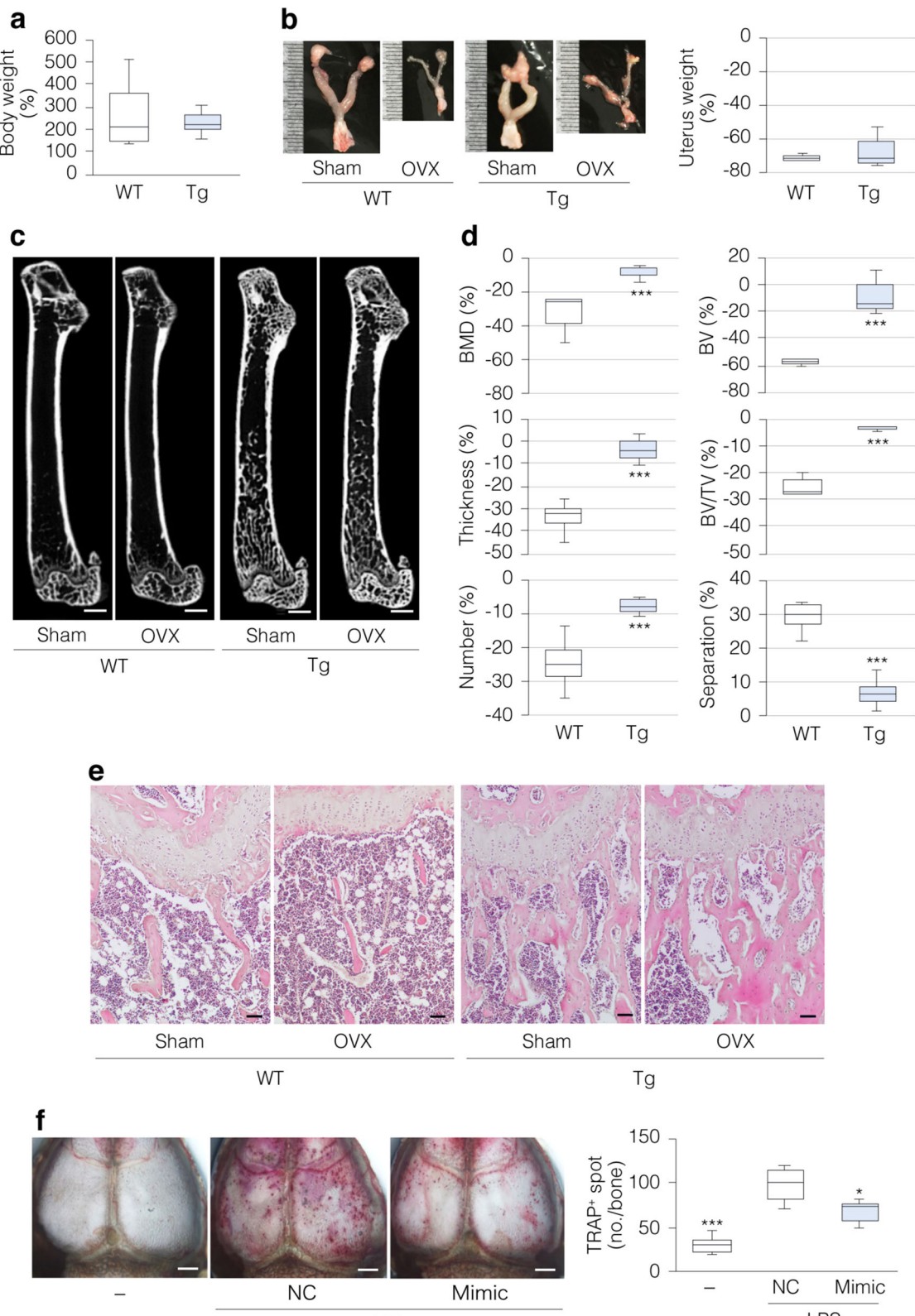

**Fig. 4 miR-125b protects mice from pathological bone loss. a–e** Eight-week-old Tg and WT female mice were either sham-operated or ovariectomized, and analyzed 4 weeks later; values (**a**, **b**, **d**) are shown as percent changes of sham-operated mice ($n = 5$) vs. OVX mice ($n = 7$). **a** Body weights. **b** Representative macroscopic images (scale bars, 1 mm) and wet weights of uteri. **c** Representative images of μCT analysis of femurs (scale bars, 1 mm). **d** Bone morphometric analysis of trabecular bone in distal femurs. Parameters based on μCT analysis were estimated as described in Fig. 2 (**d**). **e** Representative images of H&E staining of femurs (scale bars, 50 μm). **f** LPS-induced osteolysis with injections of miR-125b mimic (Mimic) or negative control miRNA (NC) over the calvariae. Representative images of whole-mount TRAP staining (scale bars, 200 μm) and quantification of TRAP⁺ spots ($n = 6$). *$P < 0.05$ and ***$P < 0.001$ vs. matched control by Student's $t$-test (**d**) and by Tukey's multiple comparison (**f**).

(RISCs)[37]. The presence of miRNAs in various body fluids is now well-established[38], but we show that the biologically active miR-125b within RISC accumulates and is retained in the extracellular bone matrix. Of established genes involved in osteoclastogenesis, we confirmed that *Prdm1* is a target of miR-125b. PRDM1 was originally isolated in human osteosarcoma cell lines[39] and acts as a transcriptional repressor. In addition to osteoclastogenesis[16,17], PRDM1 is involved in plasma cell fate determination[40], and forced expression of miR-125b decreases PRDM1 levels in the human lymphoma cell line PC-K8 via its binding activity to *Prdm1* 3′UTR[41]. Of identified targets of miR-125b, physiological silencing of miR-125b is required in mouse granulocytic differentiation[42] and B-cell development[43] in vitro via the targeting of *Jund* and *S1pr1*, respectively. The other established target of miR-125b, *Lyn28*, is involved in myeloid vs. B-cell fate determination in mouse hematopoietic stem cell cultures[44]. However, FACS analysis showed no significant differences in bone marrow cells except for a small but substantial reduction in T cells in Tg vs. WT mice. Together the data suggest that the accumulation of MVs in bone matrix creates the conditions by which osteoblast-derived miR-125b regulates osteoclastogenesis. During bone mineralization, calcium phosphate crystals grow in and break through MVs[9], resulting in miR-125b capture in mineralizing bone. As is true with other active proteins embedded in bone matrix, e.g., TGF-β1 (ref. [45]) during bone resorption, miR-125b would be released allowing its anti-osteoclastogenic activity to be manifest. This also predicts that there is a certain threshold of bone resorption required to achieve levels of miR-125b for its anti-osteoclastogenic effects.

Antiresorptive agents such as bisphosphonates and anti-RANKL antibodies have been used for many years, especially in the treatment of patients with osteoporosis and bone metastasis. These agents block bone resorption, resulting in suppression of bone turnover[46,47]. While the clinical benefits of anti-resorptive agents are clear, reduced bone turnover can have deleterious effects, e.g., medication-related osteonecrosis of the jaws, leading to ongoing efforts to identify additional approaches to maintaining and improving bone mass in a variety of conditions. It is interesting that overexpression of miR-125b in osteoblasts has a protective effect against both OVX- and NX-induced bone loss in our Tg mice and miR-125b rescues LPS-dependent osteolysis. Taken together, osteoblast–osteoclast communication via the MV cargo miR-125b represents a potential therapeutic target for bone loss (Fig. 5). As a corollary to this, future research directions must also include more detailed dissection of physiological role(s) of miR-125b in bone, as well as the mechanism(s) underlying selective accumulation of miR-125b in MVs, retention of miR-125b activity in mineralized bone overtime and delivery-uptake of miR-125b by osteoclast precursors and possibly other cells (e.g., T-cells) in the bone microenvironment.

## Methods

**Animals**. C57BL/6J and ddY mice and timed-pregnant Wistar rats were obtained from CLEA Japan or Charles River Laboratories Japan. Tg mice overexpressing miR-125b in osteoblasts were developed on the C57BL/6J background. Animal use and procedures were approved by the Institutional Animal Care and Use Committee at the Central Institute for Experimental Animals and the Committee of Animal Experimentation at Hiroshima University (#A14-078).

**Vector construction**. A 422-base-pair mouse genomic sequence including pre-miRNA-125b-1 and rabbit βglobin intron and poly A were amplified by PCR from mouse and rabbit genomic DNA, respectively, and cloned into pBlueScript II SK (+) (miR-125b vector, UNITECH)[48]. A 4072-base-pair human osteocalcin promoter[27] was amplified from human BAC clone (RP11-964F7). The human osteocalcin promoter fragment was inserted into the *NotI/SpeI* sites of the miR-125b vector, and the 5690-base-pair *NotI/SalI* fragment was isolated, purified, and used for microinjection. Tg mice were identified by PCR. Primer pairs are described in Supplementary Table 3.

**Microcomputed tomography (μCT)**. Bones were fixed in 4% paraformaldehyde (PFA) in PBS at 4 ℃ overnight and stored in 70% ethanol. Samples were imaged using Skyscan 1176 at an X-ray energy of 40 kV with a voxel size of 17.5 μm on each side and an exposure time of 230 ms (Bruker microCT). Image reconstruction and bone morphometry (proximal tibiae) were performed using NRecon and CTvox software, respectively (Bruker microCT).

**Calcein double-labeling and plastic sections**. Calcein (100 μg/kg body weight) was administered intraperitoneally twice (at a 6-day interval) to 10-week-old male mice. Femurs were dissected 2 days after the second administration, fixed in 4% PFA in PBS and embedded in Epon by standard techniques, and the MAR was determined on plastic semi-thin sections studied under fluorescence microscopy[49].

**Histology**. PFA-fixed bones were decalcified with 10% EDTA in PBS at 4 ℃ for a week and embedded in paraffin by standard techniques. Deparaffinized sections (4 μm thick) were stained with TRAP to evaluate osteoclasts[16–18]. To determine osteoblasts by ALP staining, sections were rinsed with Tris-buffered saline (20 mM Tris-HCl, 137 mM NaCl; pH 7.6) containing 0.025% Triton X-100 and incubated with Protein Block (Dako), followed by incubation with anti-ALPL antibody (×200, Proteintech) in Can Get Signal® Solution A (Toyobo) at 4 ℃ overnight. Alexa Fluor® 594-conjugated antibody (Thermo Fisher Scientific) was used as the secondary antibody. Fluoro-KEEPER with DAPI was used for nuclear staining.

**Analysis of osteoclast progenitors**. Bone marrow cells were prepared from femurs and tibiae of 10-week-old male mice. Cells were passed through a cell strainer (70 μm; Falcon®), centrifuged, and the resultant cell pellets were resuspended in ACK lysis buffer (0.15 M NH₄Cl, 0.01 M KHCO₃, and 1 mM Na₂ EDTA; pH7.4) (hemolysis). After gentle agitation for 2 min, cell suspensions were centrifuged and rinsed. Resuspended cells ($4 \times 10^5$ cells in 100 μL of 0.5% BSA in PBS) were treated with TruStain fcX for 5 min on ice, followed by incubation with labeled antibodies or isotype controls for 60 min at 4 ℃. Cells were then treated with 1 mM EDTA and 2% BSA in PBS, centrifuged and resuspended with 0.5% BSA in PBS. Osteoclast progenitors were determined by FACS (LSRFortessa X-20, BD Biosciences) using CD115-Alexa Fluor 488, CD11b-APC, and CD117-PE (BioLegend)[28] (Supplementary Table 4).

**Ovariectomy (OVX)**. Eight-week-old female mice were overiectomized or sham-operated[32], and samples were collected 4 weeks later.

**LPS-induced osteolysis in vivo**. LPS (*Escherichia coli*, 026:B65; Lot #025M4088V; Sigma-Aldrich) was administered subcutaneously over the calvariae[30] of 8-week-old male ddY mice at days 0, 1, 3, and 5 (3.75 mg/kg/day). A miR-125b mimic or its scrambled negative control (7 nmol each, Ambion) in AteloGene® (Koken) was injected subcutaneously into the region close to the LPS-treated sites 12 h before day 0 and at days 2 and 4. Calvariae were dissected at day 6 and fixed in 95% ethanol and 5% acetic acid for whole-mount TRAP staining.

**Cell cultures**. MC3T3-E1 cells and ST2 cells were purchased from the RIKEN BRC. Human osteoblastic cells were kindly provided by DV Biologics. Mouse and rat calvaria cells were obtained from newborn C57BL/6 mice (around 3 days old) and 21-day-old fetal Wistar rats, respectively[50]. Briefly, calvariae were aseptically dissected and sequentially digested five times with type I collagenase (Sigma-Aldrich) for 10–20 min. Cell fractions, except for the first one, were separately cultured in α-MEM containing 10% FBS (Corning) and pooled the next day. We also established the MC3T3-E1 cell subclone (E1^low) that expresses miR-125b at low levels by limiting dilution cloning[51]. To determine osteoblastogenesis, cells were plated at 3000 cells/cm² in the above medium additionally supplemented with 50 μg/mL ascorbic acid (osteogenic medium) with or without 10 mM β-glycerophosphate (βGP) for 2–3 weeks. Cells were fixed with 4% PFA in PBS and subjected to ALP and von Kossa staining. Image J 1.48 software (NIH) was used to measure ALP- or ALP/von Kossa-positive (ALP⁺/von Kossa⁺) areas. ALP⁺/von Kossa⁺ nodules were counted under a microscope.

Bone marrow cells from 9–12-week-old male mice were hemolyzed as above and cultured at 500,000 cells/cm² in non-adherent cell culture dishes with α-MEM supplemented with 10% FBS and 100 ng/mL M-CSF (R&D Systems) for 2 days. After removing non-adherent cells by washing with PBS, BMMs were collected using Cellstripper™ (Corning), passaged one time, and replated at 8000–20,000 cells/cm² and cultured with 50 ng/mL RANKL (Oriental Yeast) and 50 ng/mL M-CSF for 4–5 days[16,18]. RAW-D cells were plated at 2000 cells/cm² in the presence or absence of 50 or 100 ng/mL RANKL for 4 days[20]. TRAP-positive multinuclear cells (TRAP⁺ MNCs; five nuclei or more) were regarded as osteoclasts. In some resorption assays, Tg and WT BMMs (9500 cells/cm²) were cultured on calvarial bone chips (6 mm × 6 mm) for 6 days, followed by fixation with 4% PFA in PBS, TRAP staining, and Image J analysis. Calvarial bone chips were collected from either Tg or WT newborn mice and preserved in ice-cold methanol for a few days at −20 ℃. All cells were cultured at 37 ℃ in a humidified atmosphere with 5% CO₂ with medium changes every second or third day, unless otherwise specified.

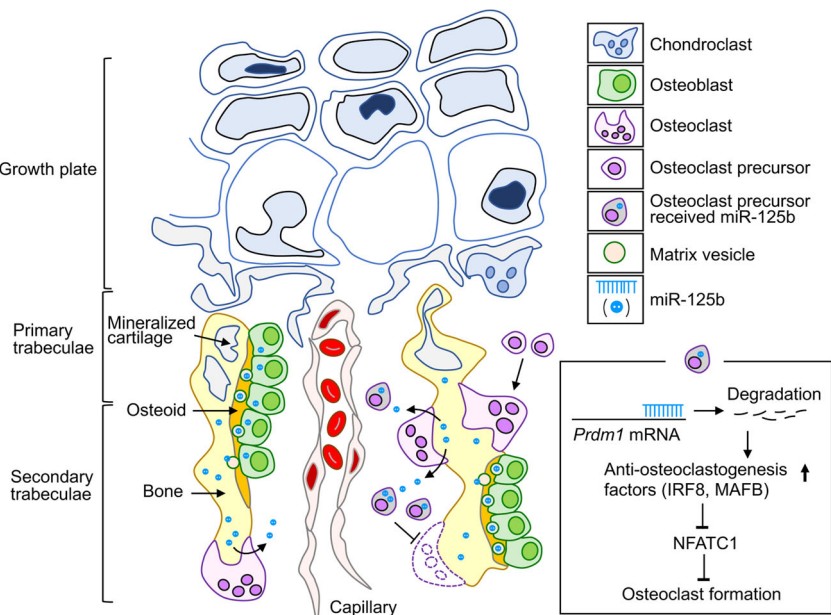

**Fig. 5 Schematic diagram of osteoblast–osteoclast communication by miR-125b.** Osteoblasts secrete matrix vesicles (MVs), including their cargo miR-125b, which accumulate in unmineralized bone matrix (osteoid). Through calcium phosphate crystal growth and MV membrane disruption, miR-125b is trapped in mineralized bone. miR-125b is taken up by osteoclast precursors such as during osteoclastic bone resorption. miR-125b in osteoclast precursors binds to *Prdm1* mRNA to decrease Prdm1 protein level, resulting in increased anti-osteoclastogenesis factors, such as IRF8 and MAFB, and thereby decreased osteoclast formation and associated bone resorption.

**Resorption pit formation assay**. BMMs at 100,000–280,000 cells on dentin slices (Wako) or bovine cortical bone slices were grown in α-MEM supplemented with 10% FCS, with 50 ng/mL M-CSF and 50 ng/mL RANKL up to 7–14 days. Dentin/bone slices were rinsed with ammonia water, stained with hematoxylin or toluidine blue[21] and resorption pit areas measured by Image J.

**Isolation of MVs**. MVs were isolated from the ECM deposited by osteoblasts[52]. Briefly, MC3T3-E1 cells, primary calvaria cells, and human osteoblastic cells were grown at 3000 cells/cm² in osteogenic medium. To confirm the formation of bone-like nodules, βGP was added to parallel cultures. When the ECM (osteoid-like nodules) formed (~3 weeks), cultures without βGP were treated with 500 units/mL collagenase (Sigma-Aldrich) at 37 °C for 45 min under static conditions and for 15 min with shaking. Samples were then centrifuged at $20,000 \times g$ for 20 min, and pellets were subjected to protein and total RNA extraction as cell extracts. Supernatants were centrifuged at $100,000 \times g$ for 60 min, and the resultant pellets were resuspended in 100 mM mannitol in 10 mM HEPES (pH 7.5), followed by centrifugation at $350,000 \times g$ for 30 min. Finally, MV pellets were resuspended in PBS and aliquots (2 μg protein/μL stock solution) were stored in liquid nitrogen until use. Two micrograms of protein/mL or less of MVs were used for bioassays.

**BCA protein assay**. Protein levels in lysates of MVs or cells (Cell-LyEX MP, TOYO B-Net) including phosphatase inhibitors) were determined using colorimetric detection according to the manufacturer's instructions (Pierce™ BCA protein assay kit, Thermo Fisher Scientific).

**Comprehensive analysis of miRNAs in MVs**. miRNAs in MVs from MC3T3-E1 cells were extracted three times independently using a mirVana™ miRNA isolation kit, according to the manufacturer's instructions (Ambion). Probe labeling (using 100 ng of total RNA) and hybridization were performed using a miRNA Complete Labeling and Hyb kit and a SurePrint G3 mouse miRNA microarray rel.17 kit, respectively (Agilent Technologies). miRNA profiling for three MVs was analyzed using an Agilent G2539A microarray scanner and analysis software GeneSpring GX (version 12.5.0, Agilent Technologies). Of miRNAs identified, we focused on those common to humans and mice, and predicted targets were searched using public databases (miRTarBase: http://mirtarbase.mbc.nctu.edu.tw; miRDB: http://mirdb.org/miRDB/; RNA22: https://cm.jefferson.edu/rna22v2/; and TargetScan-Human 6.2: http://www.targetscan.org/).

**AGO2 immunoprecipitation of bones**. The osteocyte isolation method[53] with slight modification was used to obtain miRNAs from bone matrices. Briefly, calvariae from 8–15-week-old male C57BL/6 mice were cut into small pieces (2 × 2 mm), washed, and digested with 300 U/mL collagenase (type I, Sigma-Aldrich) in isolation buffer (70 mM NaCl, 10 mM NaHCO₃, 60 mM sorbitol, 30 mM KCl, 3 mM K₂HPO₄, 1 mM CaCl₂, 1 mg/mL BSA, and 5 mg/mL glucose in 25 mM

HEPES; pH7.4) at 37 °C for 20 min with shaking. Undigested bone pieces were then stirred two times with 10% EDTA in PBS at 37 °C for 20 min and 30 min. All digestion fractions were divided into cell pellets and bone matrix extracts by centrifugation. miRNAs were trapped by antibodies against AGO2 as part of the RISC[37]. Aliquots of cell pellets lysed with lysis buffer and bone matrix extracts were incubated with anti-AGO2 mouse antibody-conjugated magnetic beads at 4 °C overnight according to the manufacturer's instructions (MagCapture miRNA kit, Wako). Mouse IgG (Santa Cruz Biotechnology)-conjugated Dynabeads Protein G (Life Technologies) were used as control. After washing, immunoprecipitates with magnetic beads were eluted and processed for RNA extraction according to the manufacturer's instructions.

**miR-125b mimic transfection**. RAW-D cells at 2000–3000 cells/cm² were grown overnight and transfected with miR-125b mimic or its scrambled negative control (10 nM each) using RNAiMAX reagent (Life Technologies) or HiPerFect Transfection Reagent (Qiagen), according to the manufacturers' instructions. After 3 h, a small amount of fresh medium including RANKL (final 50 ng/mL) was added to cultures, and cells were maintained for an additional few days.

**Prdm1 3′UTR luciferase reporter vector and luciferase assay**. The 3′UTR of *Prdm1* (2693-3145 in NCBI accession #: NM_007548.3) was amplified from genomic DNA of RAW-D cells. The amplified 3′UTR was cloned into the *SpeI* and *HindIII* sites of pMIR-REPORT (Invitrogen). The mutant 3′UTR[29] was generated using primer sets and cloned as described above. Primer sets are shown in Supplementary Table 3. RAW-D cells were grown at 25,000 cells/cm² overnight and then co-transfected with reporter plasmid (80 ng) and pGL44.74[hRluc/TK] vector (40 ng; Promega; Madison, WI) using Attractene Transfection Reagent (Qiagen), according to the manufacturer's instructions. Twenty-four hours later, miR-125b mimic or negative control (10 nM each, Ambion) was introduced into cells as described above. Cells were maintained for an additional 36 h and then lysed in passive lysis buffer (Promega). Measurements were performed using the Dual-luciferase® reporter assay system (Promega).

**PCR analysis**. Genomic DNA was extracted and the transgene was determined using KOD FX Neo (Toyobo) with specific primer sets. miRNAs in conditioned media were isolated by using a miRNeasy Serum/Plasma Advanced kit (Qiagen). Total RNA was isolated from cells using TriPure isolation reagent (Roche Diagnostics), and reverse-transcribed using ReverTra Ace® (Toyobo). For miRNAs, TaqMan® microRNA assays (Applied Biosystems), and their corresponding primer sets, including U6 (cells and MVs) and miR-16 (bone matrix) as internal controls, were used according to the manufacturers' instructions. For mRNAs, *Rpl32* served as internal control, and qPCR amplification was conducted using Fast SYBR® green master mix (Applied Biosystems). The sequences of primer sets are shown in Supplementary Table 3.

**Statistics and reproducibility**. Data are expressed as mean ± SD. Non-numerical data are shown as representative results of three independent experiments. Statistical differences were evaluated using either one-way or two-way analysis of variance (ANOVA), followed by Tukey's multiple comparison (multiple groups) or Student's $t$-test (two groups). A $P$-value < 0.05 was considered statistically significant.

**Reporting summary**. Further information on research design is available in the Nature Research Reporting Summary linked to this article.

## Data availability

The majority of the data supporting the findings of this study are included in this published article. Additionally, raw data are available from the corresponding author on reasonable request. The source data underlying plots are presented in Supplementary Data 1. The miRNA microarray data are deposited into NCBI GEO (accession number: GSE140242). miRNAs identified in MVs are shown in Supplementary Data 2.

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

## Acknowledgements

We thank S. Hiyama, M. Okumura, and A. Tomiyama for technical assistance. We also appreciate J. E. Aubin for helpful discussions and comments on the manuscript. We thank T. Kukita for the gift of RAW-D cells. T.M. and Y.T. were supported in part by MEXT KAKENHI (JP16K11443, T.M.; JP26861548, Y.T.). Y.Y. was supported by MEXT KAKENHI (JP18K19647), the Raffinee International Foundation and the Ono Pharmaceutical Foundation.

## Author contributions

T.M., Y.N. and Y.T. prepared the manuscript. T.M. conducted the in vivo and ex vivo experiments, and Y.N., T.S., and Y.T. directed the in vitro analyses. Y.N., N.S. and C.F. purified and characterized MVs and screened miRNAs. Y.I. screened the target genes and performed miRNA expression analysis. F.A. and S.I. contributed to bone morphometric analyses. A.N. and Y.S. generated Tg mice. S.N. and H.Y. supported the microarray analysis. K.T. and K.K. supported data analyses. E.B. contributed to review and editing. Y.Y. directed the project and helped to prepare the manuscript. All authors discussed the results.

## Competing interests

The authors declare no competing interests.
