## [Peer Review File · Communications Biology]

Reviewers' comments:

Reviewer #1 (Remarks to the Author):

The manuscript by Minamizaki et al. delineated an attractive role of miR-125b within osteoblast-derived bone matrix vesicles (MVs) in mediating osteoblast-to-osteoclast communication. They found that the osteoblast-derived MVs could inhibit osteoclastogenesis in vitro, and identified miR-125b, one of the miRNAs highly enriched in MVs possibly mediated the inhibitory effect via targeting Prdm1. They also found that the ovariectomy/neurectomy-induced bone loss was protected in transgenic (Tg) mice overexpressing miR-125b in osteoblasts, while LPS-induced osteolysis was rescued in mice treated with miR-125b mimics.

This is a very interesting study reporting a previously unrecognized role of MVs that the miR-125b within MVs could inhibit osteoclastogenesis. The data from in vitro study elegantly demonstrated the inhibitory effect of osteoblast-derived MVs on osteoclastogenesis, of which the mechanism involved the miR-125b-mediated suppression of Prdm1, a suppressor for several anti-osteoclastogenesis factors in osteoclast precursors. However, there is still a lack of direct evidence to confirm this MV-miR-125b-mediated mechanism in the Tg mice model, although overexpression of miR-125b in osteoblasts show protection in two disease models with bone loss. Here are some of my concerns and suggestions.

- 1) The author may also look at the changes in RANKL/OPG levels from their osteoblast culture and Tg mice to determine if manipulating miR-125b levels could affect other coupling factors that may determine osteoclastogenesis.
- 2) Please give detail information on the subclone of MC3T3-E1 with lower levels of miR-125b. Is it obtained by gene editing or microRNA inhibitor transfection?
- 3) Please give the quantity of MVs added in the osteoclast and osteoblast culture in the relevant figure legends and in the method section
- 4) Is there any effect on MVs formation or secretion, or osteoid-like nodules formation, after manipulating miR-125b? Are there more MVs formed after miR-125b Tg expression in osteoblasts, considering that MVs may contain other factors affecting osteoclastogenesis.
- 5) Bone sections with Goldner's Masson trichrome staining are recommended for the Tg mice study. Since MVs are accumulated in the unmineralized bone matrix, it would be more convincing to show that a higher miR-125b levels in the osteoid (or bone matrix) from the Tg mice comparing to WT mice.
- 6) It would be more convincing to show the uptake of osteoblast-derived MVs via osteoclasts or their precursors in vivo.
- 7) It is interesting to note that although the osteoclast formation were decreased in Tg mice, there were no differences in the osteoclast formation from BMMs and bone resorption activity in vitro between Tg mice and WT mice. It is highly recommended to test if the osteoclastogenesis of BMMs from Tg mice could be affected via direct coculture with Ob from Tg mice in vitro.
- 8) For the underlying mechanism, it seems not clear how the MVs affect osteoclastogenesis physiologically or pathologically. The author may provide a schematic diagram to illustrate mechanism details. Moreover, since osteoclasts are frequently reside at mineralized bone surface, the author should explain how could MVs accumulated in unmineralized bone regulate osteoclasts?

Reviewer #2 (Remarks to the Author):

This manuscript describes that the matrix vesicles (MVs) secreted by osteoblasts contain miR-125b that suppresses the Blimp1 (Prdm1) expression, thereby inhibiting the osteoclast differentiation. The

finding is interesting because it would unveil a novel aspect of osteoblast-osteoclast communication through secreted vesicles and miRNA. The experimental procedures conducted at here seems fine. However, unfortunately the manuscript is not written efficiently and cannot be easily understood. Following points must be improved to be re-evaluated by the journal.

Major Points:

1. Logical flows taken for strategic decision in this study was not explained well in the text. Not only for the Introduction, must the result section be revised. It should explain why the authors looked at the osteoclastogenesis at first when they started to study MV. Since many readers would first want to know the effects on osteoblasts and bone cells, it is more reasonable to first explain it. If the authors discuss a bit of contradictory results with previous reports, it would make it more reasonable to describe its effects on osteoclastogenesis.
2. It is also not clear why the authors focused on miR at first when they found that MV may send messages to the osteoclast precursors. It is more reasonable to think about membrane associated molecules such as RANKL.
3. The authors predicted that the miR-125b is the contributor to the observed effects of MVs on inhibiting osteoclastogenesis. However, the way how they could find it was not thoroughly explained in the text. The authors conducted miR-microarray analysis for the MV derived from MC3T3-E1 culture (Table 1). However it is not clear how they analyzed on it (page 20, lines 403-412). For example, what were the reference samples to be compared with?

Minor Points:

1. English text is not written efficiently. If possible, the author needs to consult with a scientific English writing expert. Many unnecessary phrases are interfering with direct reading.
2. In transgenic experiments, the expression levels of miR-125b were examined on MVs derived from mutant primary osteoblasts in culture (Supplemental Figure 7d). However, it is more important to confirm it in tissue samples.
3. It is also important to examine if the possible targets of miR-125b are exactly increased in the osteoclasts co-cultured with mutant osteoblasts, MVs, or bone matrix derived from the transgenic animals. Without proving it, current title is not accurate.
4. Also in the culture experiments using RAW-D, the authors described that a possible target of Prdm1, Bcl6 was not increased in the osteoclasts transfected with miR-125b. Why does it happen?
5. Figure 2C: Why different ages and gender of control and transgenic mice were compared with in this figure (12-wks aged male control and 10-wks old female transgenic)?
6. Figure 2g and 2h: Low magnification figures are only for mutant femur. The control femur image should be provided besides. Also, how were the ROIs (square frames) determined for TRAP-positive cell analysis?

Reviewer #3 (Remarks to the Author):

This article outlines the role of miR125b in the regulation of bone metabolism through repressing Prdm1, which represses anti-osteoclastogenesis factors.

All in all it was a well written and presented paper, and the authors competently described their results with a clear discussion.

Just a few minor points to improve the manuscript:

Page 3, lines 56-57 - hypothesis is too vague: Here, we tested the hypothesis that MVs accumulated in bone matrix play an important, previously undefined role in bone metabolism.

Page 5, lines 68-70 - statement was rather confusing: MVs inhibited osteoclasts by mouse bone marrow macrophage (BMM) cells?

Discussion - whilst it is a generally well written discussion, it would be nice to see a paragraph on the future outlook of the project. What are the next steps to the results presented?
E.g. longer term in vivo experiments focusing on the therapeutical aspects of miR125b? Other animal models?

Please revise and address these points prior to publication.

We thank the reviewers for supportive comments and address each of the issues raised as follows:

- Reviewer #1

1) The author may also look at the changes in RANKL/OPG levels from their osteoblast culture and Tg mice to determine if manipulating miR-125b levels could affect other coupling factors that may determine osteoclastogenesis.

Response:

This is an interesting point. To augment our data that MVs inhibit RANKL-induced OC formation, we have added new data to show that MVs cannot replace RANKL in OC formation assays *in vitro*, suggesting that RANKL is not a MV cargo, consistent with lack of RANKL detection in global analysis of MVs from multiple species and sources (see also <http://bioinf.xmu.edu.cn/MVsCarta/>). On the other hand, the RANK decoy receptor OPG, while not reported in osteoblast MVs, has been reported in the MVs isolated from conditioned medium of vascular smooth muscle cells (<http://bioinf.xmu.edu.cn/MVsCarta/>). We therefore tested our isolated osteoblastic MVs (2 μ g protein) but could not detect OPG, suggesting that the inhibition of osteoclastogenesis by osteoblast MVs is independent of the RANKL-OPG pathway. These results have been added to Results line 82- and Fig. 1a, b and Supplementary Fig. 1c in the revised manuscript. Of course, it remains possible that other bone coupling factors are transported in MVs, but an exhaustive testing of all species within MVs for possible effects on osteoclastogenesis is beyond the scope of the current study. Please see also our responses for related Q6 and Q7.

2) Please give detail information on the subclone of MC3T3-E1 with lower levels of miR-125b. Is it obtained by gene editing or microRNA inhibitor transfection?

Response:

We used limiting dilution cloning to obtain the MC3T3-E1 subclone that expresses lower levels of miR-125b. Please see Methods section line 364- in the revised manuscript. We cited Ref. #51 (J Bone Miner Res 10, 625-, 1995) to emphasize that subcloning the MC3T3-E1 line has been used successfully previously to isolate subclones with different phenotypic characteristics.

3) Please give the quantity of MVs added in the osteoclast and osteoblast culture in the relevant figure legends and in the method section.

Response:

We apologize for incomplete descriptions for the quantity of MVs. We have now added the information to the figure legends (Fig. 1, Fig. 3, Supplementary Fig. 1, Supplementary Fig. 2, Supplementary Fig. 3, Supplementary Fig. 4 and Supplementary Fig. 7) and Methods and Supplementary methods in the revised manuscript.

4) Is there any effect on MVs formation or secretion, or osteoid-like nodules formation, after manipulating miR-125b? Are there more MVs formed after miR-125b Tg expression in osteoblasts, considering that MVs may contain other factors affecting osteoclastogenesis.

Response:

We apologize for the incomplete description of results. As shown in Fig. 3a, bone marrow cells from Tg and WT mice exhibit the same capacity for differentiation and formation of mineralized nodules in vitro. To strengthen the finding, we have added quantification of osteoid-like nodule formation in cultures of Tg and WT calvaria cells, which also differentiated comparably. Please see Results line 143- and Supplementary Fig. 7a in the revised manuscript.

5) Bone sections with Goldner's Masson trichrome staining are recommended for the Tg mice study. Since MVs are accumulated in the unmineralized bone matrix, it would be more convincing to show that a higher miR-125b levels in the osteoid (or bone matrix) from the Tg mice comparing to WT mice.

Response:

We used multiple histochemical approaches and plastic and decalcified to try to maximize the results. As noted, significant differences were seen between Tg and WT mice assessed with microCT and decalcified paraffin sections. We performed calcein labeling with plastic sections, which limits use of other staining methods, but we did also use Villanueva staining as a surrogate for Goldner's Masson trichrome staining, which we have added to Supplementary Fig. 6c in the revised manuscript.

With respect to miR-125b: As shown in Fig. 3b, we assessed miR-125b in cells and bone matrix separately by using methods that minimize contamination of matrix with cells; this uncovered significant differences in levels of miR-125b between WT and Tg bones. We have been trying to

detect miR-125b specifically in the osteoid by in situ hybridization (ISH), but have not yet succeeded, in spite of trying different protocols. As you know, it is very difficult to detect small RNAs even in soft tissue sections, because labeling sites in microRNAs are restricted. In addition, bone sections are much more easily detach from slides during the ISH procedure. It is also much harder to activate sections while also keeping bone structures intact. In any case, we will continue to attempt to detect miRNAs by in situ hybridization, but respectfully suggest that this will need to be presented at another time, if we succeed.

6) It would be more convincing to show the uptake of osteoblast-derived MVs via osteoclasts or their precursors in vivo.

Response:

We apologize that we may not have made our points clearly enough in the original manuscript. In our initial experiments in vitro, we tested whether MVs can affect osteoclastogenesis by treating osteoclast precursors with MVs; we showed dye transfer from MVs to the precursors to make the point that MVs and their contents are accessible to and have bioactivity for osteoclasts. These results led us to identify miR-125b and design the subsequent experiments. We agree that it will be useful to show miR-125b uptake in osteoclast lineage cells in vivo. But we would respectfully suggest that it is not the MVs that we expect to be taken up, but their contents, in particular miR-125b in this case. That is, based on our data, we conclude that miR-125b is released from MVs into osteoid and mineralizing bone as calcium phosphate crystals grow through MV membranes; the miR-125b thus accumulated in bone is subsequently such as during resorption acts on osteoclast precursors. We have modified the text of Results and Discussion to make this clearer (please see below). We are also undertaking experiments to show miR-125b uptake by osteoclast precursors in vivo, but believe they are beyond the scope of the current paper. To do this, we have been generating transgenic mice expressing a reporter transgene with miR-125b response element (MRE) uniquely in osteoclast precursors. Interbreeding these mice with the miR-125 Tg mice is expected to allow us to detect miR-125b uptake in osteoclast precursors by measuring the reporter fluorescence protein levels. However, this approach will need a year or more. In the meantime, to address the reviewer's issue, we have included in the revised paper additional data from ex vivo experiments using cocultures of calvariae and BMMs. This approach was done previously to

determine whether TGF- β embedded in bone matrix can influence bone cells (Nat Med, 15(7), 757-766, 2009). We have also provided new data that miR-125b is undetectable not only in conditioned media of WT but also Tg osteoblast cultures and that miR-125b exists in bone matrix ex vivo Results line 147-, Fig 3, and Supplementary Fig. 7 in the revised manuscript.

7) It is interesting to noted that although the osteoclast formation were decreased in Tg mice, there were no differences in the osteoclast formation from BMMs and bone resorption activity in vitro between Tg mice and WT mice. It is highly recommended to test if the osteoclastogenesis of BMMs from Tg mice could be affected via direct coculture with Ob from Tg mice in vitro.

Response:

Thank you for this suggestion. We did new experiments with cocultures of BMMs and osteoblasts from Tg and WT mice, and have added new data showing that there is no difference in TRAP⁺ osteoclast formation between Tg vs WT in Results line 148- and Supplementary Fig.7f in the revised manuscript. We hope our points outlined above also clarify the mechanisms that we believe underlie these results.

8) For the underlying mechanism, it seems not clear how the MVs affect osteoclastogenesis physiologically or pathologically. The author may provide a schematic diagram to illustrate mechanism details. Moreover, since osteoclasts are frequently reside at mineralized bone surface, the author should explain how could MVs accumulated in unmineralized bone regulate osteoclasts?

Response:

Again, we apologize that we were not clear enough in our original manuscript, and we believe that the new data we have added strengthen our conclusions, as outlined in response #6 above. We have also provided a schematic diagram as Supplementary Fig. 12 in the revised manuscript. Briefly, it depicts how MVs accumulate in osteoid, and thereafter are disrupted during mineralization (please see Ref #9). Eventually, miR-125b is trapped in mineralized bone (as noted, as shown earlier for TGF- β (Nat Med 15(7), 757-766, 2009)), from which it may act such as during bone resorption.

● Reviewer #2

Major Points:

1. Logical flows taken for strategic decision in this study was not explained well in the text. Not only for the Introduction, must the result section be revised. It should explain why the authors looked at the osteoclastogenesis at first when they started to study MV. Since many readers would first want to know the effects on osteoblasts and bone cells, it is more reasonable to first explain it. If the authors discuss a bit of contradictory results with previous reports, it would make it more reasonable to describe its effects on osteoclastogenesis.

Response:

We agree that we did not provide adequate rationale, and have rewritten Introduction and Results in the revised manuscript, based on the Reviewer's suggestion.

2. It is also not clear why the authors focused on miR at first when they found that MV may send messages to the osteoclast precursors. It is more reasonable to think about membrane associated molecules such as RANKL.

Response:

We apologize for the incomplete descriptions underlying our strategy. We have re-written parts of the Results to clarify why we focused on miRNA in the revised manuscript. Please see also our response to reviewer #1 Q1 for RANKL/OPG.

3. The authors predicted that the miR-125b is the contributor to the observed effects of MVs on inhibiting osteoclastogenesis. However, the way how they could find it was not thoroughly explained in the text. The authors conducted miR-microarray analysis for the MV derived from MC3T3-E1 culture (Table 1). However it is not clear how they analyzed on it (page 20, lines 403-412). For example, what were the reference samples to be compared with?

Response:

Again, we must apologize for incomplete descriptions of the miRNA-microarray analysis. Based on the data analysis algorithm provided by the software, we first established a list of miRNAs and their relative abundance in MVs, and then did additional analyses with additional databases to narrow our list to miRNAs of greater interest and then focus on miR-125b. The strategy is now better outlined in

Results line 104-, Methods line 411- and annotation of Supplementary Table 1 in the revised manuscript.

Minor Points:

1. English text is not written efficiently. If possible, the author needs to consult with a scientific English writing expert. Many unnecessary phrases are interfering with direct reading.

Response:

We had our manuscript edited by a colleague in the bone field whose first language is English.

2. In transgenic experiments, the expression levels of miR-125b were examined on MVs derived from mutant primary osteoblasts in culture (Supplemental Figure 7d). However, it is more important to confirm it in tissue samples.

Response:

We agree and apologize for the lack of appropriate explanation of results from tissue samples shown in Figure 2. We have now provided a detailed description in Results line 127- in the revised manuscript.

3. It is also important to examine if the possible targets of miR-125b are exactly increased in the osteoclasts co-cultured with mutant osteoblasts, MVs, or bone matrix derived from the transgenic animals. Without proving it, current title is not accurate.

Response:

We agree. Please see our responses to Reviewer #1 Q1, Q6 and Q7, which we hope, together with our additional data, have strengthened the conclusions that underlie the title.

4. Also in the culture experiments using RAW-D, the authors described that a possible target of Prdm1, Bcl6 was not increased in the osteoclasts transfected with miR-125b. Why does it happen?

Response:

We now provide an explanation based on the timing of our analysis and time course of Prdm1 regulation of the genes described; Bcl6 has markedly different kinetics than the other two genes assessed. This has been clarified in Results line 178- in the revised manuscript.

5. Figure 2C: Why different ages and gender of control and transgenic mice were compared with in this figure (12-wks aged male control and 10-wks old female transgenic)?

Response:

We apologize that the Figure 2 legend was so unclear. Bones were compared from age-matched WT and Tg mice; bones were assessed from 12-week-old males and 10-week-old females based on availability of mice. Figure 2(c) legend, and Results line 125- have been modified to clarify this point in the revised manuscript.

6. Figure 2g and 2h: Low magnification figures are only for mutant femur. The control femur image should be provided besides. Also, how were the ROIs (square frames) determined for TRAP-positive cell analysis?

Response:

Images of both Tg and WT femurs at low magnification are shown in Supplementary Fig. 6. Therefore, we changed Fig. 2g to delete the femur image and substitute it with a schematic image to indicate the ROIs encompassing primary versus secondary spongiosa, as defined by established histological criteria. The Results (line 135-) sections have been revised to clarify the selection of the ROIs.

- Reviewer #3

Page 3, lines 56-57 - hypothesis is too vague: Here, we tested the hypothesis that MVs accumulated in bone matrix play an important, previously undefined role in bone metabolism.

Response:

As outlined in response to Reviewer #2 Q1, we revised both our Introduction and sections of the Results to clarify our rationale and development of the hypothesis.

Page 5, lines 68-70 - statement was rather confusing: MVs inhibited osteoclasts by mouse bone marrow macrophage (BMM) cells?

Response:

We apologize and have revised the statement for clarity (line 82- in the revised manuscript).

Discussion - whilst it is a generally well written discussion, it would be nice to see a paragraph on the future outlook of the project. What are the next steps to the results presented?

E.g. longer term in vivo experiments focusing on the therapeutical aspects of miR125b? Other animal models?

Response:

Thank you for the suggestion. We have added what we believe to be logical next steps to the end of the Discussion (line 284-).

Reviewers' comments:

Reviewer #1 (Remarks to the Author):

We appreciated the replies from the authors. It is better to revise the "title" of the article again before the publication. Because the accumulation of miR-125b was not found in the bone matrix in vivo. So, the title of article could mislead the readers. In addition, it recommended the authors to compare the miR-125b expression in the extracellular vesicle (EV) and MV from osteoblast lineage cells and their role in the osteoclastogenesis in the future studies.

Reviewer #2 (Remarks to the Author):

This manuscript revised by Dr. Yoshiki et al. improved significantly. The authors' replies to my questions were appreciated. However, I found following points are still problematic. I hope the authors would consider revising it again.

Major Points:

1. Although the authors added culture experiments utilizing miR-215b expressing Tg mouse osteoblasts, the descriptions in the text is not straight forward. At first, high level of miR-215b expression has to be confirmed (they did it but described it later in lines 147-148). Then the authors may describe about the results concerning the potential for osteogenesis, and on osteoclastogenesis. Unfortunately, the authors' experiments failed to show that calvarial osteoblasts derived from Tg mice that express high level of miR-215b does inhibit the osteoclastogenesis and bone resorption in culture (lines 153-154). Although the authors assume that matrix deposited MVs and its cargo miR-125b may not function in culture experiments (line 156), this assumption is not reasonable enough because they showed Tg MV can suppress the osteoclastogenesis of RAW-D cells (Fig 3e). The authors should clarify if the difference is just because of the accumulation of MV during culture or because of cell type specificity. Possibly an experiment utilizing bone marrow macrophages and isolated Tg MV should be done.

Minor Points:

1. The authors should explain the origin of RAW-D cell. This is not a popular cell line.
2. Page , line 162: mistyping "RWA-D" may be "RAW-D".

We thank the reviewers for supportive comments and address each of the issues raised as follows:

- Reviewer #1

We appreciated the replies from the authors. It is better to revise the “title” of the article again before the publication.

Because the accumulation of miR-125b was not found in the bone matrix *in vivo*. So, the title of article could mislead the readers. In addition, it recommended the authors to compare the miR-125b expression in the extracellular vesicle (EV) and MV from osteoblast lineage cells and their role in the osteoclastogenesis in the future studies.

Response:

We apologize that we may not have made our points clearly enough in the revised manuscript. We have measured miR-125b sequestered in the bone matrix *in vivo* (Fig. 3b) and found higher levels of miR-125b levels in the bone matrix of Tg vs. WT mice. To make this clearer, we changed the label ‘ECM’ to ‘bone matrix’ in Fig. 3b and its legend. We have also added additional information to and changed the position of the corresponding section of the Materials and Methods (titled ‘AGO2 immunoprecipitation of bones’) that explains how we performed the assay. Please see also our responses to Major points of the reviewer #2. Based on this, we believe the that the original title is appropriate and respectfully suggest that it be kept.

We also thank the reviewer for the suggestion of comparing miR-125b expression in EVs and MVs of osteoblast lineage cells and assessing their role in osteoclastogenesis in future.

● Reviewer #2

Major Points:

1. Although the authors added culture experiments utilizing miR-215b expressing Tg mouse osteoblasts, the descriptions in the text is not straight forward. At first, high level of miR-215b expression has to be confirmed (they did it but described it later in lines 147-148).

Then the authors may describe about the results concerning the potential for osteogenesis, and on osteoclastogenesis. Unfortunately, the authors' experiments failed to show that calvarial osteoblasts derived from Tg mice that express high level of miR-215b does inhibit the osteoclastogenesis and bone resorption in culture (lines 153-154). Although the authors assume that matrix deposited MVs and its cargo miR-125b may not function in culture experiments (line 156), this assumption is not reasonable enough because they showed Tg MV can suppress the osteoclastogenesis of RAW-D cells (Fig 3e). The authors should clarify if the difference is just because of the accumulation of MV during culture or because of cell type specificity. Possibly an experiment utilizing bone marrow macrophages and isolated Tg MV should be done.

Response:

We apologize for the lack of clarity in presenting our results from cell and tissue samples shown in Figure 2a, b and originally explained in lines 127-128. We have re-written the text to make results clearer in lines 127-129 in the 2nd revised version.

For ex vivo experiments, again we apologize for the lack of clarity and have revised the text and figures, with inclusion of new data, to address the following points more clearly. First, as the reviewer notes, our data on direct cocultures of osteoblasts and bone marrow macrophages (BMMs) indicate that Tg osteoblasts overexpressing miR-125b do not by themselves have the ability to suppress osteoclastogenesis. While the coculture system used is suitable for assessing direct cell-cell interactions, it does not allow assessment of factors/processes requiring actual bone resorption. Second, we have assessed the ability of BMMs to differentiate into osteoclasts with resultant bone resorption (Fig. 3c) and the number of osteoclast progenitors (Fig. 3d and Supplementary Fig. 7 e) in Tg mice and found them to be equal to those in WT mice, indicating that there are no intrinsic defects in ability of Tg osteoclast precursors to form osteoclasts or in Tg osteoclast activity. Third, we do detect an inhibitory effect on osteoclastogenesis when either WT or Tg BMMs are cultured on Tg bone chips but not on WT bone chips (Fig. 3f). We have also confirmed that miR-125b accumulates in matrix vesicles and in bone matrix in vivo (Fig. 3b) but not in conditioned media from osteoblast

(calvaria cell) cultures (Supplementary Fig. 7d). These results indicate that osteoblast MV-derived miR-125b is mostly sequestered in the bone matrix. Fourth, osteoclasts continue to form in BMM cultures for 7 days, with the average lifespan of those formed being three days or less in this culture model. We have now added new data (Fig. 3f-h) in the 2nd revised version of our manuscript to show that Tg and WT BMM cultures on Tg and WT bones differentiate equally into TRAP⁺ osteoclastic cells over the initial 3 days. However, when cultured for 6 days, fewer TRAP⁺ osteoclastic cells formed on Tg vs. WT bone chips, concomitant with higher miR-125b levels detectable in conditioned media in the cultures on Tg bone (Fig. 3g in the 2nd revised version). Consistent with this, TRAP⁺ osteoclastic cell formation by RAW-D cells was inhibited, when treated with conditioned media from BMM cultures on Tg but not WT bone chips (Fig. 3h in the 2nd revised version). These results are in keeping with other now well-established results in which it has been shown that certain growth factors, e.g. TGF β , sequestered in the bone matrix can be released during bone resorption and act on osteoblastic cells (Ref. #46).

Taken together, our data suggest that miR-125b is released from bone matrix during osteoclastic bone resorption and accumulates to a level that inhibits subsequent osteoclastogenesis. To make this clear, we have re-written the second and third paragraphs (lines 156-) in the section 'miR-125b targets Prdm1 in osteoclast precursors.'

Minor Points:

1. The authors should explain the origin of RAW-D cell. This is not a popular cell line.

Response:

We agree and have mentioned the origin of RAW-D cell (line 89) with reference #22 in the 2nd revised version.

2. Page , line 162: mistyping "RWA-D" may be "RAW-D".

Response:

We apologize for the mistyping, which has been corrected.

REVIEWERS' COMMENTS:

Reviewer #1 (Remarks to the Author):

My concerns have been addressed.

Reviewer #2 (Remarks to the Author):

The secondly revised version of this manuscript improved significantly by newly added data by Figure 3g, and I feel it is sufficient to be accepted by the Journal.